# Ultrafast high-harmonic nanoscopy of magnetization dynamics

Sergey Zayko [1,2✉], Ofer Kfir [1,2,4], Michael Heigl [3], Michael Lohmann[1], Murat Sivis[1,2], Manfred Albrecht [3] & Claus Ropers [1,2]

Light-induced magnetization changes, such as all-optical switching, skyrmion nucleation, and intersite spin transfer, unfold on temporal and spatial scales down to femtoseconds and nanometers, respectively. Pump-probe spectroscopy and diffraction studies indicate that spatio-temporal dynamics may drastically affect the non-equilibrium magnetic evolution. Yet, direct real-space magnetic imaging on the relevant timescales has remained challenging. Here, we demonstrate ultrafast high-harmonic nanoscopy employing circularly polarized high-harmonic radiation for real-space imaging of femtosecond magnetization dynamics. We map quenched magnetic domains and localized spin structures in Co/Pd multilayers with a sub-wavelength spatial resolution down to 16 nm, and strobosocopically trace the local magnetization dynamics with 40 fs temporal resolution. Our compact experimental setup demonstrates the highest spatio-temporal resolution of magneto-optical imaging to date. Facilitating ultrafast imaging with high sensitivity to chiral and linear dichroism, we envisage a wide range of applications spanning magnetism, phase transitions, and carrier dynamics.

[1] 4th Physical Institute—Solids and Nanostructures, University of Göttingen, 37077 Göttingen, Germany. [2] Max Planck Institute for Biophysical Chemistry, 37077 Göttingen, Germany. [3] Institute of Physics, University of Augsburg, 86159 Augsburg, Germany. [4] Present address: School of Electrical Engineering, Tel Aviv University, 69978 Tel Aviv, Israel. ✉email: szayko@gwdg.de

Localized magnetic textures are essential elements of hard-drives, magnetic random-access memories[1], novel storage schemes[2–4], and potential building blocks for next-generation logic devices[4–7]. Topological spin objects and domain textures, in particular, attract considerable attention as topological excitations that can be nucleated, erased and controllably translated, equivalent to the writing and register-shifting of logical bits[3,4,7,8]. Due to the inherently localized nature of such magnetic features, their coupling with the environment is governed by nanoscale heterogeneity, while exchange and spin-orbit interactions define the intrinsic femtosecond time scale of magnetic dynamics. Whereas ultrafast demagnetization has been studied for decades[9], recent experiments based on extreme-UV and soft-X-ray scattering suggest a coupling between spatial magnetic properties and the temporal scale of the demagnetization process[10–13]. Similarly, magneto-transport phenomena involve an ultrafast spatio-temporal response, such as super-diffusive spin-currents[14,15], hot electrons[16], or the spin Seebeck effect[17,18], ranging down to a few tens of femtoseconds[19–21].

Access to combined nm-fs spatio-temporal resolution, as required for real-space observations of spin dynamics at the fundamental limits, has remained a considerable experimental challenge. Time-resolved measurement schemes were developed to access magnetic dynamics with sub-100 nm spatial and picosecond temporal resolutions. These are based on established imaging techniques, such as spin-polarized scanning tunneling microscopy[22], Lorentz-contrast electron microscopy[23–29], scanning transmission x-ray microscopy[30,31], high-resolution ptychography[32,33] and tomography[34,35], as well as full-field magneto-optical imaging approaches including Fourier transform holography (FTH)[31,36–38] and coherent diffractive imaging (CDI)[39,40]. Magneto-optical Kerr microscopy[41] and photoemission electron microscopy[42], on the other hand, offer femtosecond snapshots of magnetic textures, albeit at somewhat lower spatial resolution. To date, the 2014 pioneering work by von Korff–Schmising and co-workers[43] remains the only experimental demonstration of ultrafast magnetization dynamics imaged with sub-100 nm spatial and 100-fs-temporal resolution, using FTH at a free-electron laser facility. Thus, exploring the frontiers of ultrafast nanoscale magnetism still requires more widely accessible experimental techniques.

In this work, we introduce ultrafast dichroic nanoscopy based on high-harmonic radiation, reaching unprecedented spatio-temporal resolution in full-field magneto-optical imaging. Exploiting the X-ray magnetic circular dichroism (XMCD) of cobalt in Co/Pd multilayer structures, we trace the ultrafast response of nanoscale magnetic textures to femtosecond laser excitation. We capture laser-induced spin textures, ranging from micrometer-scale domains to ~80 nm-sized magnetic bubbles, with a sub-wavelength spatial resolution down to 16 nm. Ultrafast movies of nanoscale spin dynamics are recorded with a temporal resolution of 40 fs.

## Results and discussion

The experimental principle is depicted in Fig. 1a. Femtosecond pulses from a Ti:Sapphire laser amplifier (central wavelength of 800 nm, pulse energy up to 3.5 mJ, 35 fs pulse duration) are split into a pump and a probe arm. The pump pulses optically excite the sample, and circularly polarized radiation form high-harmonic generation (HHG) in a gas cell serves as the femtosecond probe for magnetic imaging. The 38th harmonic order (wavelength of ~21 nm), which provides for XMCD contrast near the M-edge of cobalt, is selected and focused onto the sample by a toroidal grating monochromator, whereas the remaining harmonic orders are blocked. The diffraction pattern of the radiation

transmitted through the sample is collected by a charge-coupled device (CCD) camera placed a few centimeters downstream. The positioning of the camera determines the collected scattering angle and the theoretical (diffraction) limit of the spatial resolution. Real-space images as a function of delay time are retrieved algorithmically by an iterative phasing of the recorded far-field intensities (Fig. 1a), based on a combination of FTH and coherent diffractive imaging. Guiding the CDI algorithm with FTH drastically improves the image recovery[44–48] and allows us to significantly increase the spatial frequencies up to which the phases are accurately reconstructed[44]. Both the dichroic phase (Fig. 1b) and absorption yield image contrast. Their relative contribution can be tuned through the choice of the probing harmonic order near the absorption edge[49,50]. Imaging of reversible magnetization dynamics is conducted stroboscopically at 1 kHz repetition rate, allowing for an evaluation of spatially-resolved magnetization dynamics (Fig. 1b, c).

The magnetic sample exhibiting an out-of-plane easy axis of magnetization is a multilayer structure of cobalt and palladium (Co/Pd) deposited on membranes made of silicon-nitride ($Si_3N_4$) or silicon (see Methods). The backside of the sample is coated with gold (180-200 nm thick) which is opaque for the high-harmonic radiation. Fields of view (FOV) with a diameter of 3.1 µm are formed by removing the gold using focused ion beam (FIB) etching. Additionally, an array of holes is milled through the entire sample thickness including the substrate and the magnetic film (c.f. Figure 2a). The array is designed such that the strong auxiliary field emanating from these holes covers the entire detector and interferometrically enhances the weak magneto-optical scattering signal from the FOV aperture[51]. This signal-enhancement scheme drastically improves the dynamic range and reduces the required dose by more than one order of magnitude compared to a conventional FTH or CDI approach.

Two of the holes (diameters of 200-400 nm) are well-separated from the FOV, to allow for a direct recovery of low-resolution holographic information that facilitates subsequent high-resolution iterative phase retrieval.

Extracting pure magnetic information requires the joint phase retrieval (phasing) of the far-field intensities recorded with left- and right-handed circularly polarized illumination. An image recorded with a single helicity is dominated by nonmagnetic contrast, characterized by a higher magnitude of the transmission and a phase offset (see Fig. 2b, c) in the holes compared to the film within the circular FOV. Only weak variations in the FOV are observed for single-helicity reconstructions, including fringes near the perimeter from wave-guiding and edge-diffraction effects[52,53]. Some contrast arises from thickness variations and contaminations of the specimen and the substrate. Illustrating the importance of a dichroic reconstruction, Fig. 2d (top) shows the complex amplitudes of all pixels in the single-helicity exit waves within the circular FOV, exhibiting considerable scatter in amplitude and phase. The dichroic signal, however, obtained by a pixel-by-pixel ratio of the complex amplitudes (R/L), shows a very well-defined double-lobed histogram distribution (Fig. 2d, bottom), reflecting the quasi-binary contrast in the final dichroic image (phase contrast in Fig. 2e). The phase of the ratio of the two complex-valued reconstructions (R/L) evidently corresponds to the XMCD-phase difference $\phi_R - \phi_L$. Similarly, the absolute value of R/L yields the dimensionless XMCD-absorption.

We have found this dichroic microscopy approach to be very robust and have recorded hundreds of magnetization maps for different samples, material systems, masks, and numerical apertures in the detection. Figure 3 shows a set of dichroic phase-contrast images of magnetic textures in Co/Pd multilayers (see Methods) with half-pitch spatial resolutions between 27 and 16 nm. Note that these are direct reconstructions without filtering

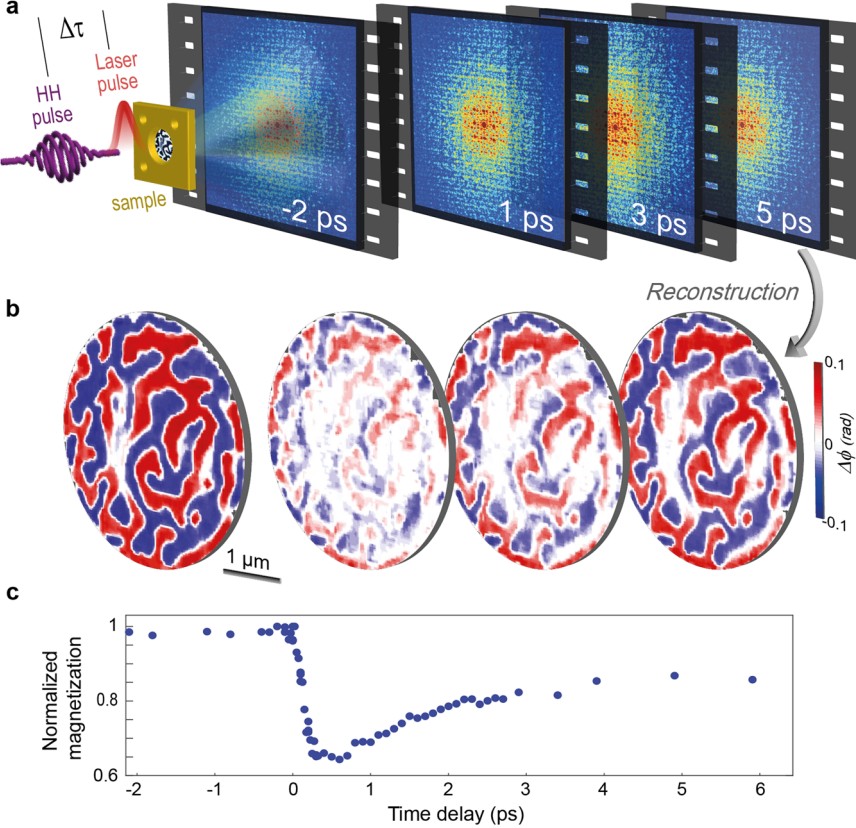

**Fig. 1 Ultrafast high-harmonic nanoscopy. a** A magnetic sample is excited with a femtosecond laser pulse and probed with a circularly polarized high-harmonic pulse (wavelength of 21 nm) at a variable delay $\Delta\tau$. For each time delay between the pump and probe pulses, a quantitative real-space image is reconstructed from the diffraction pattern of the high-harmonic beam. **b** The ratio of images obtained from opposite HHG helicities provides for a map of the magnetic contrast isolated from nonmagnetic contributions, i.e., femtosecond snapshot of the spin structures at a given delay time between pump and probe pulses. **c** Plot of the spatially-averaged normalized magnetization within the field of view as a function of delay. Each data point on the plot corresponds to an average magnetization measured from a single real-space magnetic image at a given time delay.

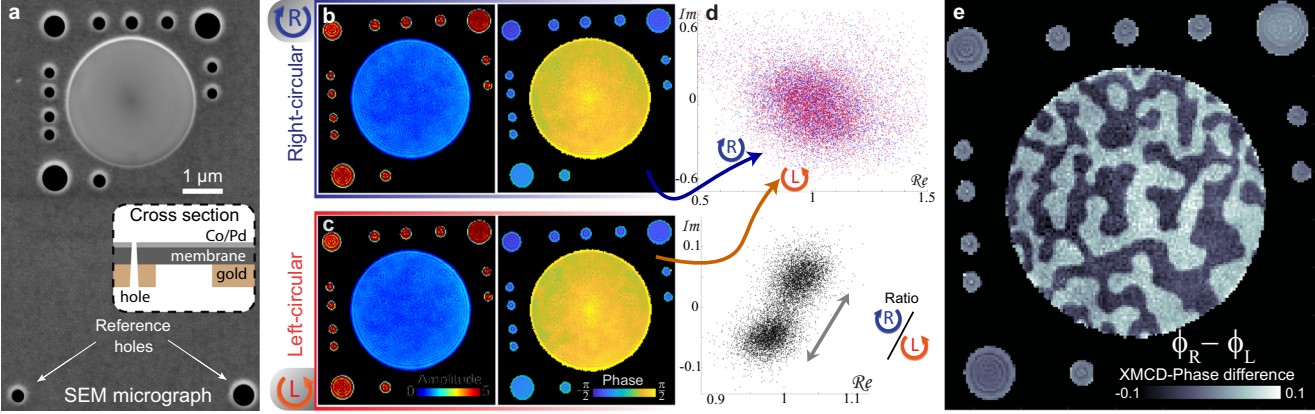

**Fig. 2 Dichroic imaging employing coherent signal enhancement. a** Scanning electron microscope (SEM) image of the sample showing a circular 3.1 μm field of view (FOV), auxiliary holes for signal enhancement and two distant reference holes (bottom) for low resolution Fourier-transform holography (see methods). Inset: Sample cross section. **b**, **c** Exit-field magnitude and phase obtained with right- (top panel) and left-handed (bottom panel) circularly polarized high-harmonic beams. Contrast from the XMCD effect (X-ray magnetic circular dichroism) is small compared to the strong non-magnetic signal. The complex amplitudes are normalized to the mean in the FOV. **d** Scatter plot of pixel-by-pixel complex exit-wave amplitudes within the circular FOV aperture (top) and their ratio(bottom). The dichroic signal is highlighted with a gray arrow of length 0.11. **e** Image of the dichroic phase-contrast, which is the difference between the phase maps presented in panel **c**.

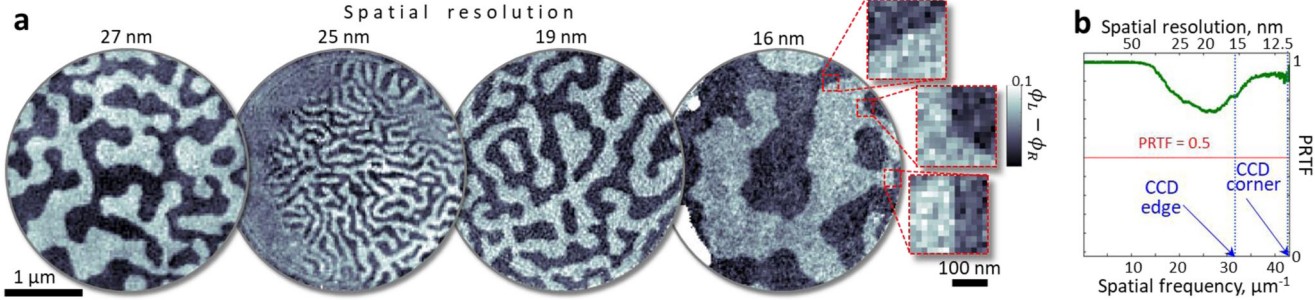

**Fig. 3 High-resolution full-field imaging of various samples with magnetic domains ranging from sub-30 nm to few μm in size. a** Magnetic circular dichroism phase-contrast images with spatial resolution of 27, 25, 19 and 16 nm (see Methods). Raw experimental data, i.e., reconstructions without image processing. **b** The phase-retrieval transfer function (PRTF) computed for the diffraction data recorded at the highest numerical aperture. The PRTF plot indicates high consistency of the retrieved phases up to the edges and corners of the CCD sensor, corresponding to spatial resolutions of 16 nm and 12 nm, respectively. The diameter of the circular field of view is 3.1 μm for all data sets.

or any pixel interpolation. We consistently reach a diffraction-limited single-pixel resolution in each of the scattering geometries used. In real space, the diffraction-limited resolution is evident in sharp contrast changes at domain walls that span a single pixel (see insets in Fig. 3a). In a far-field analysis, the resolution is evaluated using the phase retrieval transfer function (PRTF)[54,55], an objective measure for the reliability of phase retrieval procedures. The PRTF quantifies the azimuthally integrated consistency of the retrieved phases, and the conservative criterion for the resolution estimate is the inverse of the lowest spatial frequency with PRTF < 0.5. The PRTF values, as shown in Fig. 3b, stay above the 0.5 mark, indicating confidence in the retrieved phases even at the edges and the corners of the CCD sensor, corresponding to spatial frequencies (half-pitch resolutions) of 16 nm and 12 nm, respectively. Despite the general complexity of coherent diffractive imaging, our approach is implemented in such a way to allow for in-situ imaging with only few-second delays between the raw data acquisition and the subsequent image reconstruction, providing for real-time feedback during experimental sessions.

The temporal resolution of our HHG-based microscope for lensless magnetic imaging is governed by the 35-fs duration of the near-infrared pump laser, with negligible contribution from the sub-10 fs duration of the high-harmonic probe pulses (see Methods). Figure 4a, b show example XMCD phase-contrast frames of the spin dynamics movies for two different samples and fluences, reconstructed with a spatial resolution of 40 nm. The pump fluences were chosen to induce considerable demagnetization while minimizing irreversible changes of the domain pattern. Comparing consecutive frames in Fig. 4a, b, the most pronounced observation is the overall reduction of the magnetization immediately after the optical pump ($\Delta\tau = 0$) with a well-resolved timescale of 200 fs, and a subsequent partial recovery over a few picoseconds. Figure 4f compares demagnetization curves retrieved from ultrafast movies recorded at different pump fluences. Each data point on the plots represents the spatially averaged magnetization of an image at a given delay time between pump and probe pulses. Here, we use the absolute value of the magnetization to account for the two domain orientations within the FOV. Ultrafast demagnetization occurs at a constant rate, which is evident from the alignment of the demagnetization slopes for different pump fluences (Fig. 4f). The constant demagnetization rate results in longer times to reach the maximum suppression for higher pump fluences[56].

Figure 4c, d show high-fidelity reconstructions of a domain pattern before the pump and near the maximum suppression of the magnetization. The change of the normalized magnetization is displayed in Fig. 4e, revealing a standing-wave-like pattern with

features separated by about half the optical wavelength, and suggesting considerable scattering of the pump beam at the mask edges[43,57]. In a local subwavelength hotspot in the right of the FOV, a significant suppression of the magnetization down to about 20% of the initial value is observed. Segmentations in areas of different local fluence (see color-coded intervals beside the color map of Fig. 4e) yield de- and re-magnetization dynamics that strongly vary across the FOV (Fig. 4g), and which, accordingly, also differ in shape from the spatially-averaged dynamics evaluated for the same incidence fluence in Fig. 4f.

We have analyzed the time-resolved images searching for transverse spin dynamics. Despite the highest spatio-temporal resolution available to date, our data do not show unambiguous evidence for a significant broadening of domain walls, as may have been expected from the characteristics of superdiffusive spin currents and previous spatially averaging experiments[10–13,56,58]. Several reasons may account for this observation. First, the current spatial resolution is close to the estimated domain wall width for this system, and spin transport processes may be limited to the scale of only a few nanometers, as also suggested by a very recent diffraction study[59]. Second, a noticeable broadening of the domain wall may appear for a substantially higher fluence, which is consistent with the data provided in Ref. [11]. However, we observe that at pump fluences for which the local magnetization is suppressed by about 50% or more, the real-space domain structure starts to exhibit irreversible changes before any noticeable DW broadening occurs. In other words, domain wall broadening appears to be coupled to irreversible changes of the domain structure. We identify the fluence threshold for irreversible changes of the domain structure through a local reduction of contrast appearing at negative delay times, i.e., a millisecond after the previous pulse (see Supplementary Movies employing various excitation fluences). In the presence of irreversible changes of the domain texture, the diffraction pattern is composed of an incoherent sum over all configurations within the integration time. Nonetheless, the CDI algorithm consistently reconstructs the parts of the domain pattern outside of the hotspot regions, which are not affected by pulse-to-pulse fluctuations.

Whereas such irreversible changes cannot be transiently probed stroboscopically, we can obtain insights into the final magnetic state produced using single or multiple pulses at higher fluences, an approach which has previously been employed to generate topological magnetic features, including vortex-antivortex networks or skyrmions[28,38,60–62]. Figure 5 shows a sequence of qualitatively different final magnetic states induced by high-fluence laser excitation. The images show characteristic spin textures that were obtained by an abrupt interruption of the excitation by blocking the laser beam, resulting in a quench across

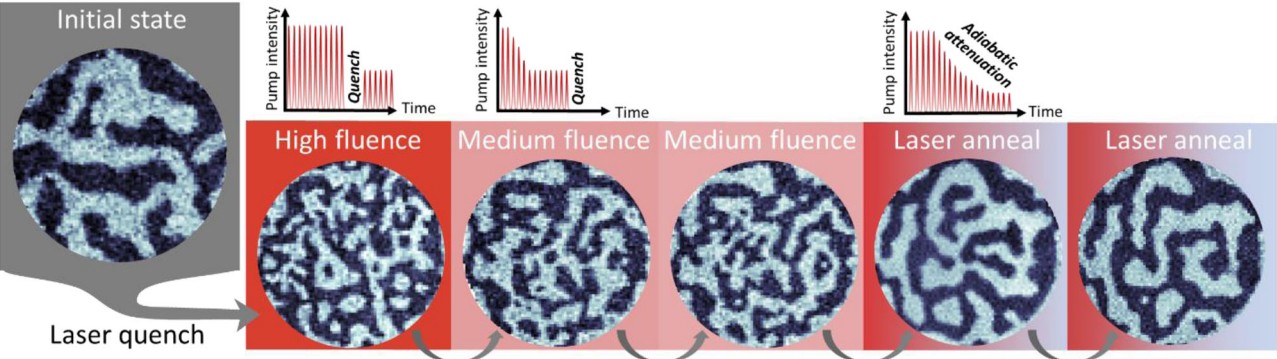

**Fig. 4 Ultrafast imaging of femtosecond spin dynamics. a**, **b** Example frames from movies of magnetization dynamics for two samples driven at different fluences (experimental data as reconstructed). **c** Interpolated averaged images over 4 frames at negative delays, i.e., before the excitation. Data from panel **b**. **d** Image after pumping (four frames around 1 ps). Note the strong local variations in the suppression of the magnetization. **e** The demagnetization map is the ratio of the magnetic-contrast before and after the pump, highlighting regions with stronger demagnetization. **f** Evolution of the average magnetization (absolute value of the dichroic images) within the field of view(FOV), for different incident pump fluences. Incident fluences 1, 2, and 3 on the structure amount to 1.2(2), 1.3(1) and 1.4(1) mJ/cm$^2$, respectively. **g** Local magnetization dynamics for the given incident laser fluence in different regions of the sample. The regions are selected from panel **e**, as indicated beside the color bar.

**Fig. 5 Laser-induced manipulation of the spin texture.** The magnetic domain arrangements of the same sample can be drastically modified by the laser excitation. A laser quench from high fluence (4.0(6) mJ/cm$^2$) leads to the formation of nanoscale magnetic structures and a higher density of domain walls compared to a quench from medium pump fluence (1.4(1) mJ/cm$^2$). A gradual reduction of the incident pump fluence (from 4.0(6) mJ/cm$^2$ to zero) allows for the recovery of a magnetic pattern with large µm scale domains. The diameter of the circular field of view is 3.1 µm.

the ferromagnetic phase transition. Depending on the excitation parameters, the resulting magnetic structures vary from large micrometer-scale domains to domain-wall-rich textures. In particular, quenching from high fluence leads to a rather fractured pattern of small domains, while intermediate fluences produce more regular domains hosting individual sparse bubbles (see Fig. 5). Continuously reducing the excitation from high to low fluence eliminates these bubbles, leaving the structure with comparatively large and smooth domains. This in-situ preparation of different magnetic textures may be combined with ultrafast imaging under sub-threshold excitation to study the possible influence of domain sizes and domain-wall correlation lengths on spatiotemporal demagnetization dynamics and spin currents. Finally, enhancing the reproducible re-magnetization of a particular domain state necessary for stroboscopic measurements may be achieved with more controlled local excitation, for example, using nanostructures for tailoring localized optical fields with higher spatial gradients.

In conclusion, we have developed ultrafast high-harmonic dichroic nanoscopy for robust full-field imaging of magnetic spin textures and their femtosecond evolution on the nanoscale. These results represent both the highest spatial and the highest spatio-temporal resolution in full-field magneto-optical imaging obtained to date, irrespective of the wavelength, technique or the radiation source used. Reaching the diffraction limit with a high-coherence source, our microscope employs a wavelength one to two orders of magnitude longer than typically required for sub-50 nm resolution. We envisage a wide range of applications for high-harmonic nanoscopy using linear or circular dichroism, with many further applications in ultrafast and element-specific spintronic imaging, but also extending to the tracking of hot carrier populations, structural and electronic phase transitions, and chemical transformations.

## Methods

**HHG source.** High-harmonic up-conversion is driven with a bi-circular two-color field tailored using a MAZEL-TOV apparatus from a linearly polarized laser beam with the central wavelength at 800 nm. We optimize the generation conditions to improve the harmonic yield near the maximum of the magneto-optical phase signal of the cobalt M-edge[49,50]. The generated harmonic spectrum is dispersed with a toroidal grating and the 38th harmonic (~21 nm wavelength) is isolated with a slit before it enters the imaging chamber. Although we find magnetic contrast also for harmonic orders 37th, 38th, 39th, and 40th, the maximum dichroic contrast is at the 38th harmonic order and it comprises mostly phase contrast. Thus, the images throughout the paper present the XMCD-phase of the 38th harmonic (with an exception within Fig. 2). Although not directly used in this work, we note that the suppressed 39th harmonic can be accessed using fine-tuning of the driving optical fields, as we show in Ref. [63].

We use a diffraction grating to remove adjacent harmonics and ensure sufficient temporal coherence across the mask for high-resolution lensless imaging[64,65]. Moreover, the spatial coherence from the bichromatic circular driver is higher compared to a conventional HHG scheme with linearly polarized excitation due to a reduced number of allowed electron trajectories[63,66]. The observation of a full suppression of every 3rd harmonic order is a robust and reliable indication for generation conditions corresponding to circularly polarized harmonics. In order to maximize the magneto-optical signal, the circular polarization has to be provided at the sample position. This can be achieved by adjusting the waveplate angle of the MAZEL-TOV to pre-compensate the polarization dependent reflection of the optical elements between the sample and the generation point, in this case the diffraction grating. To optimize and verify the circular polarization at the sample plane, we use an extreme-UV polarization analyzer based on nanoscale slit arrangement that allows for in-situ polarization measurement in a single acquisition[67].

**Imaging scheme.** We employ holographically guided lensless imaging, in which the sample mask contains a circular FOV aperture, two or more holographic reference holes, and an array of auxiliary holes. The masks are designed according to the following criteria:

1. A confinement of the exit wave to fulfill the sampling requirement for phase retrieval[64]. This criterion defines the maximum size of the FOV aperture, which in our case was successfully tested for circular apertures with diameters of 2, 3, 4, 5, and 6 μm.

2. Sufficiently small holographic reference holes (diameters between 200 and 400 nm) to provide for exact information on the sample geometry and the real-space support via FTH directly from the measured data. Having two reference holes makes the automatic support deconvolution process more straightforward.

3. An array of additional auxiliary holes of a specific shape and arrangement is introduced in the vicinity of the sample to ensure homogeneous coverage of the detector with a strong scattering signal that interferes with the weak magneto-optical signal from the FOV aperture and coherently enhances it above the detector noise level. This array reduces the required dose by more than an order of magnitude, depending on the numerical aperture used. Furthermore, the dynamic range of the scattering data becomes lower which simplifies data collection and reconstruction.

For every delay frame in the pump-probe movie, we record two sets of diffraction data using a left- and right-handed circularly polarized harmonic illumination. Each data set is composed of 10–20 diffraction patterns with 3–30 s exposure time (depending on the mask geometry and NA used) to increase SNR and the dynamic range of the diffraction data. In the case of high-resolution imaging experiments shown in Fig. 3, a total exposure time of up to 15 min was required for sub-wavelength spatial resolution reconstructions. To further improve the SNR for these data sets, we acquire short and long exposure time diffraction patterns that are subsequently merged into a single HDR data set. Starting from a random first guess, deconvolved holographic support and using a modified RAAR algorithm[68], we first reconstruct a data set with left-handed circular polarized probe using 130–150 iterations and afterwards reconstruct the data set of the opposite helicity. If both data sets are accurately reconstructed and precisely aligned, the ratio of these two independently reconstructed exit wave amplitudes of opposite helicities eliminate non-dichroic contributions, providing for a pure magnetic phase and absorption contrast image. To accelerate the iterative process to just 10 iterations and to skip the sub-pixel alignment procedure, the reconstruction for the opposite helicity (as well as for the all data sets of a given time series) can be initiated from a previously obtained reconstruction. Importantly, such an approach mitigates possible phase retrieval and imaging artefacts for diffraction data with poor SNR, low oversampling ratio, or insufficient coherence properties of the probe. Irrespective of the initial random guess, the dichroic reconstructions are virtually identical (as evident form the PRTF plotted in Fig. 2c), thus no averaging of several reconstructions or filtering is required for diffraction data with high SNR. Furthermore, no additional phase retrieval constraints have to be imposed, and we attribute this to the improved spatial and temporal coherence of the developed HHG source as well as the coherent signal enhancement mechanism provided by the auxiliary reference holes.

We note that time-invariant regions of the sample, i.e., the auxiliary and reference holes, can be enforced as an additional real-space constraint in the reconstruction process to potentially relax the SNR requirement for each delay frame, following alternative approaches specifically designed for pump-probe diffraction data[69].

**Resolution estimate.** To estimate the spatial resolution, we first measure the accuracy of the phase retrieval process in the far-field using a well-established phase retrieval transfer function (PRTF) method[70]. Figure 3b shows the PRTF plot for 20 individual reconstructions initiated from a random first guess demonstrating a consistent phase retrieval throughout the entire momentum space (diffraction limited imaging system). The spatial resolution in this case is the highest spatial frequency recorded. Additionally, we verify the spatial resolution in real space as the smallest resolvable feature in the reconstruction. We find multiple magnetic domains with sub-25 nm transverse sizes that are clearly resolved. However, the smallest dimensions in the investigated samples are magnetic domain walls that are expected to be in the range between 10 and 20 nm[71]. The reconstructed images exhibit multiple domain walls appearing as a single-pixel step as evident in Fig. 3. For material systems with larger domain walls, for example, studied in[12,13] the demonstrated resolution will be sufficient to resolve domain walls within several real-space peixels.

The temporal resolution of our approach is conservatively estimated as 40 fs using an optical cross-correlation of the 35 fs pump beam (verified with Spectral Phase Interferometry for Direct Electric-field Reconstruction) and expected sub-10 fs harmonic probe. First, we measure the spatio-temporal shear introduced by the diffraction grating to the probe pulse. Second, we compute a fraction of this shear incident onto a 3 μm large FOV of the object. Finally, we consider the non-collinear excitation on the FOV and measure the convolution of the resulting pulses.

**Magnetic film fabrication.** The magnetic Co/Pd ML films used for time-resolved data have the following structure: [Pd(0.75 nm)/Co(0.55 nm)]$_9$. For the samples shown in Fig. 3a the Co layer thickness were varied from 0.4 to 0.55 nm. All samples exhibit strong perpendicular magnetic anisotropy with an easy axis magnetization pointing out-of-the film plane[72,73]. The MLs were prepared at RT by dc magnetron sputtering from elemental targets on silicon and silicone-nitride membranes. The sputter process was carried out using an Ar working pressure of $3.5 \times 10^{-3}$ mbar in an ultra-high vacuum chamber (base pressure <$10^{-8}$ mbar). For all films, 1.5 nm of Cr and 2.0 nm of Pd were used as seed layer. To protect the

films from corrosion 2 nm of Pd were used as cover layer. The thicknesses of the layers were estimated from the areal densities measured by a calibrated quartz balance during deposition. For the Co/Pd composition used, a stack of nine bilayers gives the optimal magneto-optical contrast at the used wavelength. However, the approach can also be applied for other sample thicknesses, and stacks in the range between 2 and 22 Co/Pd bilayers are expected to produce a contrast at least half of that shown here. The contrast suppression for thicker samples is dominated by absorption in Pd. For a pure cobalt foil (fully magnetized parallel to the HHG beam (38th order) at a normal incidence angle), this would correspond to a thickness range between 1 nm and about 120 nm.

## Data availability

The data supporting the findings of this study are available within the paper and its Supplementary Information files and from the corresponding authors upon reasonable request.

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

## Acknowledgements

We gratefully acknowledge the support and insightful discussions with Tim Salditt, Marcell Möller, Thomas Danz, Tobias Heinrich, Philipp Buchsteiner. O.K. gratefully acknowledges funding from the European Union's Horizon 2020 research and innovation programme under the Marie Skłodowska-Curie grant agreement No. 752533. This work was funded by the Deutsche Forschungsgemeinschaft (DFG) in the Collaborative Research Center "Nanoscale Photonic Imaging" (DFG-SFB 755, project C08). S.Z acknowledges funding from the Campus Laboratory for Advanced Imaging, Microscopy and Spectroscopy (AIMS) at the University of Göttingen.

## Author contributions

S.Z., O.K. and C.R. conceived and designed the experiment with contributions from M.L. and M.S. The samples were designed by S.Z. O.K., M.S., M.H. and M.A., and fabricated by M.H., M.S. and M.A. Measurements and data analysis were performed by S.Z. and O.K. The manuscript was written by S.Z., O.K. and C.R. with contributions from all authors.

## Funding

## Competing interests

The authors declare no competing interests.
