## [Peer Review File · Nature Communications]

Reviewers' Comments:

Reviewer #1:

Remarks to the Author:

The manuscript by Zakyo et al describes the use of holographic imaging with high-harmonic radiation of the temporal evolution of magnetic stripe domains in CoPd samples following femtosecond laser excitation. The authors use a combination of Fourier transform holography and coherent diffractive imaging to achieve high spatial resolution of 16-27nm. This instrumental advance is certainly impressive and justifies publication. However, I feel it is more suitable for a journal specializing in instrumentation advances. Scientifically the authors essentially report the well known attenuation of the domain magnetization following femtosecond laser excitation. Also the influence of plasmonic modification of the pump laser has already been reported. The lack of observed domain wall broadening reported with diffraction techniques is assigned to irreversible domain changes that cannot be imaged in the present experiment. To recommend publication in Nature Communications I would like to see a compelling scientific result beyond what has already been reported with such time-resolved imaging techniques.

Reviewer #2:

Remarks to the Author:

The authors report on the first fs time-resolved imaging of magnetic nanostructures using XUV radiation from a high-harmonic source. The samples investigated are prototypical nanoscale domains in a Co/Pd multilayer. The ultrafast mechanism investigated is ultrafast demagnetization using a structured pump beam. While this experiment is conceptually very similar to the work of von Korff Schmising et al. 2014, the achievements of present study go significantly beyond the previous results. The progress in spatial and temporal resolution as well as in the overall image quality and photon and time efficiency make this work a new milestone in the field. Moreover, the study stands out by the use of a laboratory XUV source instead of FEL or synchrotron-radiation sources typically used for short-wavelength magnetic-contrast imaging. The method and apparatus employed for (static) imaging have already been introduced in a previous publication of the same groups (Kfir et al. 2017). In the present study, the spatial resolution for static images was even further improved and has now closed the gap to the best results achieved with standard synchrotron-based imaging methods (e.g., STXM, ptychography, TXM, holography). The experimental finding of the absence of domain-wall broadening by transversal spin currents as previously proposed adds important limits to strength of the effect. Similar doubts have also recently published by Weder et al. 2020 (DOI: 10.1063/4.0000017).

The work is original and the experimental results are significant. The literature is comprehensively cited. Without any doubt, Nature Communications would be an appropriate journal for a publication of these excellent and pioneering results. However, I would like to ask the authors to answer and consider the following remarks:

1. The setup is (and probably will be for a long time) limited to XUV radiation. While the spatial resolution is impressively reduced to the wavelength limit, I still have concerns with respect to the sample thickness. The XMCD at the transition-metal M-edges is roughly an order of magnitude smaller compared to the L-edges. At the same time, the absorption length is much shorter at long wavelengths leading to a strong dilemma between contrast and transmission. Where do the authors see the application window for this wavelength? How limited in terms of samples is the method?

2. The authors stress several times that the signal from the additional apertures coherently amplifies the magnetic scattering and helps to improve the SNR. This idea was experimentally first discussed in Flewett et al 2012 (DOI: 10.1364/OE.20.029210). It has been previously shown that the shot noise cannot be improved by this method (Shintake 2010 DOI: 10.1103/PhysRevE.81.019901). I would assume that the improvement observed in the present study is related to the small single-photon signal on the CCD compared to the noise level in this wavelength regime. For higher photon energies enabling single-photon counting, I would not expect such a prominent effect. Do the authors agree?

3. I find the paragraph on the "laser-manipulation" scientifically pointless. The author do not give any meaningful experimental details and the results are postponed to a later publication. The paragraph and corresponding figure read like a news outlet rather than a scientific report. This part should be omitted or meaningfully extended.

4. I find the presentation of Fig. 2e over-artistic. In this way it does not have any scientific meaning. All scales are missing and the data can hardly be discerned. In general, I find the Figs (in particular Fig. 1 and 2) very densely packed which seems to be unnecessary for Nature Communications. The authors should check all figures for missing scalebars, colorbars and labels.

Reviewer #3:

Remarks to the Author:

The authors report on a new technique to image in real space magnetic nanostructures with very good spatio-temporal resolution. For that, they use a combination of the Fourier Transform Holography and Coherent Diffractive Imaging techniques to retrieve more easily the real space image from the collected diffraction patterns. Those diffraction patterns are measured thanks to a high harmonic generation source providing photons circularly polarized at Co M edge (21 nm, 59 eV). This tabletop technique allows them to achieve a spatial resolution of tens of nanometer with a time resolution of tens of femtosecond, as they demonstrate on a Co/Pd multilayer. Their results highlight the potential of this technique to challenge remaining femtomagnetism questions about magneto-transport properties at ultrafast time scales.

While there is no doubt that the research has been carried out thoroughly since the experimental data are of high quality and the analysis appears to be sound, some information and considerations should be addressed more clearly in the text.

1. In the experiment explanation part of their article and Figure 2, the authors talk about dichroic phase contrast and get the magnetic domains from the dichroic ratio of circular left (CL) and right (CR) polarization, while in the results part and Figure 3 and 4 the authors use the XMCD phase contrast which is the difference between CL and CR polarization. It is not clear why they use different 'XMCD' normalization to get the magnetic contrast. As far as I know usually people look at the difference between CL and CR and not the ratio.

2. The authors claim that they can be sensitive to the magnitude and phase of the exit field by tuning the wavelength of the probe close to the absorption edge. While I agree with that, they neither gave the probing wavelength they use in figure 2 b and c for the amplitude and phase imaging, nor how they performed such fine wavelength tuning with HHG.

3. Although the article does demonstrate a spatial resolution of 50 nm, it is not clear enough in my opinion, that the 16nm resolution claimed is really achieved. This number is retrieved from the phase-retrieval transfer function which is not explained (in this article but also the reference referred to). In Figure 3 b, the authors show very nice pictures with a field of view corresponding to approximately 2 μm of diameter. This does not show at all that we can see a small magnetic object such as bubbles or grains smaller than 50 nm as it is claimed for the '16 nm' spatial resolution. Having a zoom of this picture with a scale bar of 50 nm will more definitively convince the reader of a sub 50nm spatial resolution.

4. In their results (Figure 4) the authors find a maximum of demagnetization between 500 fs to 1 ps. This is quite slow for Co/Pd multilayer. Could the authors give an explanation or hypothesis about this 'slower' time compared to the already existing literature? Furthermore, the author claims that they do not see any broadening of the domain wall, but even from a static picture, it is not clear that we can see those domain walls. From pictures 4 a and b, it looks like the domain wall are really sharp, while from pictures 4 c and d, the white line separating both domains look like they have the resolution to indeed see the domain wall. Is it just a color artifact? To compare domain walls dynamic, could the authors look at a line cut comparison of the before and after pump picture? Finally, the authors do not give information about the static size of the magnetic domains and domain walls. It seems that the domain size is of the order of 150 nm which is quite large compare to the domain wall size (<10 nm probably). Additionally to the hypothesis of a not strong enough pump, the authors could also argue about the size ratio between magnetic domains and domain wall not big enough. Indeed the previous study of Hennes et al and Zusin et al the domain walls are quite large (~40 nm).

Finally, please find below minor comments:

1. Line 77, the authors define Coherent Diffractive Imaging as CDI but it is already done earlier in the text
2. Line 92, the authors claim that the signal-enhancement scheme allows a reduction of the required dose, but it is not clear to which dose they refer to. The number of incoming photons?
3. In the HHG source part of the Methods section, the authors say that they measure in situ polarization measurement, but they do not give any percentage of the circular polarization.
4. Ref. 56 is now published on PRB (<https://doi.org/10.1103/PhysRevB.102.174437>)

We would like to thank the Reviewers for the careful reviews of our manuscript and the constructive feedback.

We have addressed questions and comments raised by the Reviewers and give a point-by-point response below. *The Reviewers' comments are styled as italicized blue* while our responses use the standard coloring and font.

Reviewer #1:

The manuscript by Zaky et al describes the use of holographic imaging with high-harmonic radiation of the temporal evolution of magnetic stripe domains in CoPd samples following femtosecond laser excitation. The authors use a combination of Fourier transform holography and coherent diffractive imaging to achieve high spatial resolution of 16-27nm. This instrumental advance is certainly impressive and justifies publication. However, I feel it is more suitable for a journal specializing in instrumentation advances. Scientifically the authors essentially report the well known attenuation of the domain magnetization following femtosecond laser excitation. Also the influence of plasmonic modification of the pump laser has already been reported. The lack of observed domain wall broadening reported with diffraction techniques is assigned to irreversible domain changes that cannot be imaged in the present experiment. To recommend publication in Nature Communications I would like to see a compelling scientific result beyond what has already been reported with such time-resolved imaging techniques.

We thank the Reviewer for expressing his/her appreciation of the high quality of our imaging results. We acknowledge the Reviewer's perspective on the scientific merits and broad relevance of our paper, which we would like to address in two parts.

On the methodical or technical side, we would like to reiterate that to date, there appears to be only a single work on ultrafast magneto-optical imaging at comparable length scales [43]. That pioneering experiment was conducted at a bright free-electron laser (FEL) facility – a rare instrument which exceeds our compact (HHG) setup in investment, infrastructure dimensions and running cost by multiple orders of magnitude. We would argue that in most scientific contexts, achieving and surpassing some technological ability at such a fraction of the instrumental effort and cost would be regarded a scientific breakthrough. Moreover, considering almost an order of magnitude improvement in spatiotemporal resolution over previously available capabilities, we believe our work represents a major scientific advance. Given the accessibility of HHG setups and the coherence and pulse duration they can offer, our work will be of interest for a broad scientific community as a novel tool for studying a wide range of ultrafast phenomena, also beyond magnetism.

Concerning the physics of domain wall broadening: Our approach targets ultrafast spin dynamics in real space, providing information that is inaccessible from spatially-averaged scattering and spectroscopy experiments. We respectfully disagree with the Reviewer's assertion that our results primarily confirm previous observations, and we are not aware of previous results which would allow to draw the same conclusions as our study. Featuring the highest spatio-temporal resolution available to date, the present data show that despite considerable local demagnetization, magnetic domain wall widths in a prototypical material remain unaffected and exhibit no transient broadening on these length scales. Moreover, the projected increase of domain wall widths obtained from previous scattering experiments require fluences

significantly higher than those inducing irreversible changes of the domain structure. In other words, domain wall broadening appears to be coupled to irreversible changes of the domain structure. This observation has not been made before, and it has required a real-space characterization of the kind introduced here. We acknowledge that these points were not communicated clearly enough in the original submission, and we have revised the manuscript accordingly. Besides further emphasizing the methodological advance of the work, we have revised part of the discussion to now read:

“We have analyzed the time-resolved images searching for transverse spin dynamics. Despite the highest spatio-temporal resolution available to date, our data do not show unambiguous evidence for a significant broadening of domain walls, as may have been expected from the characteristics of superdiffusive spin currents and previous spatially averaging experiments [10–13,56,58]. Several reasons may account for this observation. First, the domain wall broadening may be significantly smaller than previously thought, as also suggested by the very recent study on spin dynamics in Co/Pd multilayers using far-field analysis and spatially modulated laser excitation [59]. Second, a noticeable broadening of the domain wall may appear for a substantially higher fluence, which is consistent with the data provided in Ref. [11]. However, we observe that at pump fluences for which the local magnetization is suppressed by about 50 % or more, the real-space domain structure starts to exhibit irreversible changes before any noticeable DW broadening occurs. In other words, domain wall broadening appears to be coupled to irreversible changes of the domain structure. We identify the fluence threshold for irreversible changes of the domain structure through a local reduction of contrast appearing at negative delay times, i.e., a millisecond after the previous pulse (see supplementary movies employing various excitation fluences).”

Moreover, in response to comments made by Reviewer#2, we have revised the discussion of the single-pulse domain modifications and have placed it near the end of the manuscript, relating to the irreversible changes observed. We believe that these results now connect more naturally to the time-resolved data and illustrate both the capabilities of our setup and the overall physical boundaries discovered for reversible domain wall broadening.

We would like to thank the Reviewer again for the helpful comments, and hope the Reviewer finds the manuscript improved in the present form.

Reviewer #2

The authors report on the first fs time-resolved imaging of magnetic nanostructures using XUV radiation from a high-harmonic source. The samples investigated are prototypical nanoscale domains in a Co/Pd multilayer. The ultrafast mechanism investigated is ultrafast demagnetization using a structured pump beam. While this experiment is conceptually very similar to the work of von Korff Schmising et al. 2014, the achievements of present study go significantly beyond the previous results. The progress in spatial and temporal resolution as well as in the overall image quality and photon and time efficiency make this work a new milestone in the field. Moreover, the study stands out by the use of a laboratory XUV source instead of FEL or synchrotron-radiation sources typically used for short-wavelength magnetic-contrast imaging. The method and apparatus employed for (static) imaging have already been introduced in a previous publication of the same groups (Kfir et al. 2017). In the present study, the spatial resolution for static images was even further improved and has now closed the gap to the best results achieved with standard synchrotron-based imaging methods (e.g., STXM, ptychography, TXM, holography). The experimental finding of the absence of domain-wall broadening by transversal spin currents as previously proposed adds important limits to strength of the effect. Similar doubts have also recently published by Weder et al. 2020 (DOI: 10.1063/4.0000017).

The work is original and the experimental results are significant. The literature is comprehensively cited. Without any doubt, Nature Communications would be an appropriate journal for a publication of these excellent and pioneering results. However, I would like to ask the authors to answer and consider the following remarks:

We thank the Reviewer for placing this work in a broader context, and for the explicit support for publication in Nature Communications.

1. The setup is (and probably will be for a long time) limited to XUV radiation. While the spatial resolution is impressively reduced to the wavelength limit, I still have concerns with respect to the sample thickness. The XMCD at the transition-metal M-edges is roughly an order of magnitude smaller compared to the L-edges. At the same time, the absorption length is much shorter at long wavelengths leading to a strong dilemma between contrast and transmission. Where do the author see the application window for this wavelength? How limited in terms of samples is the method?

We thank the Reviewer for this comment. Although XMCD imaging with HHG could of course reach the keV photon energy range, let us address these issues for our particular system. Indeed, the magnetic contrast and the total absorption of the sample define the accessible range of sample thicknesses, depending on the specific composition. We estimate that sufficient image contrast will be obtained for a relatively broad range of sample thicknesses. For example, for similar Co/Pd bilayers, image contrast will reach at least half of the imaging contrast presented in this work (the actual number of bilayers used here is 9) using the stacking sequence between 2 and 22 bilayers (assuming that magnetization anisotropy stays the same). For pure Cobalt, we estimate a suitable thickness range from about 1 nm up to 120 nm, and even thicker for less absorptive materials such as Iron.

In order to address the question of suitable sample thickness in the manuscript, we added the following statements to the section that describe the sample preparation in the Methods: *“For the Co/Pd composition used, a stack of 9 bilayers gives the optimal magneto-optical contrast at the used wavelength. However, the approach can also be applied for other sample thicknesses, and stacks in the range between 2 and 22 Co/Pd bilayers are expected to produce a contrast at least half of that shown here. The contrast suppression for thicker samples is dominated by absorption in Pd. For a pure cobalt foil (fully magnetized parallel to the HHG beam (38th order) at normal incidence angle), this would correspond to a thickness range between 1 nm and about 120 nm. “*

2. The authors stress several times that the signal from the additional apertures coherently amplifies the magnetic scattering and helps to improve the SNR. This idea was experimentally first discussed in Flewett et al 2012 (DOI: 10.1364/OE.20.029210). It has been previously shown that the shot noise cannot be improved by this method (Shintake 2010 DOI: 10.1103/PhysRevE.81.019901). I would assume that the improvement observed in the present study is related to the small single-photon signal on the CCD compared to the noise level in this wavelength regime. For higher photon energies enabling single-photon counting, I would not expect such a prominent effect. Do the authors agree?

We thank the Reviewer for bringing up this important point. In our case, the readout and dark noise are indeed comparable to the single-photon signal, and hence, utilization of an auxiliary fields is vital, as it lifts the signal substantially above the instrumental noise. Because the phase-retrieval algorithm is nonlinear, the contribution of this enhanced SNR to the image quality is disproportionately advantageous compared to the linear imaging operator used, for example, in FTH or STXM. Besides the instrumental noise aspect, the signal from the auxiliary reference holes improves the dynamic range of the diffraction signal, which simplifies data detection as well as the reconstruction process. Using our approach, we expect a substantial improvement of the CDI image quality also for the data recorded at much higher photon energies.

To address this question, we have added the reference given by the Reviewer and now summarize these aspects in the Methods to read as: *“An array of additional auxiliary holes of a specific shape and arrangement is introduced in the vicinity of the sample to ensure homogeneous coverage of the detector with a strong scattering signal that interferes with the weak magneto-optical signal from the FOV aperture and coherently enhances it above the detector noise level. This array reduces the required dose by more than an order of magnitude, depending on the numerical aperture used. Furthermore, the dynamic range of the scattering data becomes lower, which simplifies data collection and reconstruction.”*

3. I find the paragraph on the “laser-manipulation” scientifically pointless. The author do not give any meaningful experimental details and the results are postponed to a later publication. The paragraph and corresponding figure read like a news outlet rather than a scientific report. This part should be omitted or meaningfully extended.

We thank the Reviewer for this comment. In response, we decided to revise the corresponding discussion and place a separate figure at the end of the manuscript. Since no evidence of domain wall softening (within the resolution limits of our microscope) was observed for moderate pump fluences, we believe that the data with irreversible changes of the spin textures at higher pump fluences serves as a logical extension of our observations. Moreover, the observed qualitative modification of the magnetic domain structure at

different pump fluences may give another perspective on the interpretation of scattering data without immediate access to real-space information. Indeed, differences in transient domain structures may contribute to differences in previous not spatially resolved observations, in which the role of the domain walls to the demagnetization dynamics was investigated, for example, in Moisan et al [56] and Vodungbo et al [10].

4. I find the presentation of Fig. 2e over-artistic. In this way it does not have any scientific meaning. All scales are missing and the data can hardly be discerned. In general, I find the Figs (in particular Fig. 1 and 2) very densely packed which seems to be unnecessary for Nature Communications. The authors should check all figures for missing scalebars, colorbars and labels.

We thank the Reviewer for this comment. We removed panel Fig. 2e in the revised manuscript. Moreover, in response to the comments from Reviewers #2 and #3, we further modified Fig. 3.

We thank the Reviewer again for the comments and suggestions, and we hope the Reviewer finds the manuscript improved in its present form.

Reviewer #3

Their results highlight the potential of this technique to challenge remaining femtomagnetism questions about magneto-transport properties at ultrafast time scales.

We thank the Reviewer for the positive feedback and for highlighting the relevance of our work.

1. In the experiment explanation part of their article and Figure 2, the authors talk about dichroic phase contrast and get the magnetic domains from the dichroic ratio of circular left (CL) and right (CR) polarization, while in the results part and Figure 3 and 4 the authors use the XMCD phase contrast which is the difference between CL and CR polarization. It is not clear why they use different 'XMCD' normalization to get the magnetic contrast. As far as I know usually people look at the difference between CL and CR and not the ratio.

We thank the Reviewer for this comment. To clarify, we use the ratio of the complex-valued reconstructions from opposite helicities (CL /CR) as the dichroic signal. The difference between the phases of CL and CR is thus the phase of the ratio (CL/CR). We acknowledge that this point required clarification in the manuscript. The revised manuscript now states: **“The phase of the ratio of the two complex-valued reconstructions (L/R) maps the XMCD-phase difference $\phi_L - \phi_R$. Similarly, the absolute value of L/R maps the unit-less XMCD-absorption”**. We also added a sentence to Fig. 1: **“The ratio of images obtained from opposite HHG helicities provides for a map of the magnetic contrast isolated from non-magnetic contributions...”**.

2. The authors claim that they can be sensitive to the magnitude and phase of the exit field by tuning the wavelength of the probe close to the absorption edge. While I agree with that, they neither gave the probing wavelength they use in figure 2 b and c for the amplitude and phase imaging, nor how they performed such fine wavelength tuning with HHG.

We thank the Reviewer for pointing out that the wording in the submitted manuscript was somewhat ambiguous. The modified sentences now reads as: "Both the dichroic phase and absorption yield image contrast. Their relative contribution and the overall dichroic contrast can be tuned by the choice of the probing harmonic order near the absorption edge [48,49]"

Imaging with different harmonics, we observe magnetic contrast for the 37, 38, 39 and 40th harmonic order of the laser. The maximum dichroic contrast is at the 38th harmonic order, and it comprises mostly phase contrast. Thus, the images throughout the paper present the XMCD-phase of the 38th harmonic. We now also add this information to the methods which reads: "The generated harmonic spectrum is dispersed with a toroidal grating and the 38th harmonic (~21 nm wavelength) is isolated with a slit before it enters the imaging chamber. Although we find magnetic contrast also for harmonic orders 37th, 38th, 39th and 40th, the maximum dichroic contrast is at the 38th harmonic order and it comprises mostly phase contrast. Thus, the images throughout the paper present the XMCD-phase of the 38th harmonic (with an exception within Fig. 2). Although not directly used in this work, we note that the suppressed 39th harmonic can be accessed using fine-tuning of the driving optical fields, as we show in Ref. [62]"

3. Although the article does demonstrate a spatial resolution of 50 nm, it is not clear enough in my opinion, that the 16nm resolution claimed is really achieved. This number is retrieved from the phase-retrieval transfer function which is not explained (in this article but also the reference referred to). In Figure 3 b, the authors show very nice pictures with a field of view corresponding to approximately 2 um of diameter. This does not show at all that we can see a small magnetic object such as bubbles or grains smaller than 50 nm as it is claimed for the '16 nm' spatial resolution. Having a zoom of this picture with a scale bar of 50 nm will more definitively convince the reader of a sub 50nm spatial resolution.

We thank the Reviewer for this suggestion. We revised both the text and figure for clarification. First, we increased the apparent size of the images and added zoom-in segments to highlight the smallest features (domain walls) in Fig. 3, which show multiple reproducible single-pixel steps in contrast. We have decided against showing interpolated data for these transitions, to stay true to the actual physical pixel size of the reconstruction which was 16 nm for these data sets. Second, we added clarifications to the meaning of the PRTF in the manuscript, which is a quantitative resolution measure: "In a far-field analysis, the resolution is evaluated using the phase retrieval transfer function (PRTF) [54,55], an objective measure for the reliability of phase retrieval procedures. The PRTF quantifies the azimuthally integrated consistency of the retrieved phases, and the conservative criterion for the resolution estimate is the inverse of the lowest spatial frequency with $PRTF < 0.5$. The PRTF values, as shown in Fig. 3b, stay above the 0.5 mark, indicating confidence in the retrieved phases even at the edges and the corners of the CCD sensor, corresponding to spatial frequencies (half-pitch resolutions) of 16 nm and 12 nm, respectively." Third, we added the abovementioned PRTF references to the methods section as well.

4. In their results (Figure 4) the authors find a maximum of demagnetization between 500 fs to 1 ps. This is quite slow for Co/Pd multilayer. Could the authors give an explanation or hypothesis about this 'slower' time compared to the already existing literature?

We observe that the laser-induced demagnetization occurs at a nearly constant rate, which implies that the time difference between the incidence of the pump pulse and the time of lowest magnetization is fluence-dependent. For the pump fluences used, we observed that this time ranges from about 200 to 700 fs, which is consistent with the existing literature [9,10,13,15,19,20,56]. Following the Reviewer's comment, we clarify this aspect in the revised manuscript, to now read: "Ultrafast demagnetization occurs at a constant rate, which is evident from the alignment of the demagnetization slopes for different pump fluences (Fig. 4f). The constant demagnetization rate results in longer times to reach the maximum suppression for higher pump fluences [56]." Finally, we revised the plots in Figure 4 to more clearly show the region of interest.

Furthermore, the author claims that they do not see any broadening of the domain wall, but even from a static picture, it is not clear that we can see those domain walls. From pictures 4 a and b, it looks like the domain wall are really sharp, while from pictures 4 c and d, the white line separating both domains look like they have the resolution to indeed see the domain wall. Is it just a color artifact? To compare domain walls dynamic, could the authors look at a line cut comparison of the before and after pump picture?

We thank the Reviewer for this comment. The images on Fig.4 a,b are reconstructed directly from the experimental data for the noted time frames. The experimental noise is currently the limiting factor for determining the domain-wall dynamics with high statistical significance using a single line-out. For example, frame-varying white noise in near the magnetization flipping lines could (mistakenly) be interpreted as dynamics. To improve the visibility of the domains and their walls, each of the panels, c and d, is averaged over 4 consequent delay frames at negative times and 4 consequent frames after the excitation, respectively. Additionally, we performed an interpolation to decrease the pixel size. We now add the following line to the figure caption to clarify this issue: "**b Example frames from movies of magnetization dynamics for two samples driven at different fluences (experimental data as reconstructed). c Interpolated averaged images over 4 frames at negative delays, i.e., before the optical pump hits the sample**". The colormap for Fig. 4 c,d was chosen to highlight the regions with magnetization transition, i.e., domain walls, so that a small increase of the domain wall widths will be directly evident from the image. This manifest itself, for example, on the DW width near the right edge of the sample which appears to be increasing in size due to highly localized excitation along the domain wall caused by a plasmonic field enhancement on the edge.

Finally, the authors do not give information about the static size of the magnetic domains and domain walls. It seems that the domain size is of the order of 150 nm which is quite large compare to the domain wall size (<10 nm probably). Additionally to the hypothesis of a not strong enough pump, the authors could also argue about the size ratio between magnetic domains and domain wall not big enough. Indeed the previous study of Hennes et al and Zusin et al the domain walls are quite large (~40 nm).

Following the Reviewer's suggestion, in the revised manuscript, we added scale bars and descriptions to all figures for a direct comparison of the domain sizes on each data set, whereas the estimation of domain wall widths is now given in the methods section: "**magnetic domain walls that are expected to be in the range between 10 and 20 nm[71]**". Figure 5 shows that for the same sample, the possible metastable spin arrangements can be drastically different. Moreover, their shapes and sizes heavily depend on the excitation parameters, thus also affecting the ratio between the domain walls and domain sizes.

While the ratio of DW width to domain width is relevant in diffraction, we believe that it should not affect whether or not DW broadening is resolved in real-space imaging. Our study shows that there is no significant DW softening for well-separated domain walls and the spatial resolution given. It will be interesting to study systems with different domain textures, including dense DW networks for which accelerated demagnetization has previously been reported [10].

Finally, please find below minor comments:

1.Line 77, the authors define Coherent Diffractive Imaging as CDI but it is already done earlier in the text

2.Line 92, the authors claim that the signal-enhancement scheme allows a reduction of the required dose, but it is not clear to which dose they refer to. The number of incoming photons?

3.In the HHG source part of the Methods section, the authors say that they measure in situ polarization measurement, but they do not give any percentage of the circular polarization.

4.Ref. 56 is now published on PRB (<https://doi.org/10.1103/PhysRevB.102.174437>)

We thank the Reviewer noting these points, and we fixed them all in the revision. We have now clarified the statement about the in-situ polarization measurement. In the day-to-day operation, we straightforwardly rely on the high degree of circular polarization guaranteed by the HHG selection rule. The employed EUV-polarization analyzer allows to fine-tune the polarization to account for the polarization sensitive beam-transport in order to increase the degree of circular polarization to 1 at the sample.

We thank the Reviewer again for the constructive criticism and hope the Reviewer finds the manuscript improved in its present form.

Reviewers' Comments:

Reviewer #1:

Remarks to the Author:

The authors have revised their manuscript to address my concerns. In particular they identify the pump fluence dependent crossover to irreversible changes in the recorded domain pattern. This is an important result that sets the present investigation apart from results using spatially averaged scattering techniques and provides valuable new scientific insight. I therefore support publication of the manuscript in its present form.

Reviewer #2:

Remarks to the Author:

The authors have addressed my comments suitably and I find the manuscript significantly improved. In particular, the part on static modifications of the domain patterns is now well integrated into the presentation of the findings. I still recommend publication in Nature Communications due to the seminal methodical progress combined with important physical findings.

I only have some minor issues remaining that should be addressed before publication:

1. In Fig. 2 d, the meaning of the double-headed arrow is not explained. Should it give a scale between the top and bottom panel? As the bottom panel shows the ratio (rather than the difference), this would not make sense mathematically, in my opinion. Which data points are shown in Fig 2d? Only the ones inside the main FOV? Do the color scales in Fig 2 b,c agree with the axis scaling in panel d top? The caption of Fig 2 still explains panel e as f. More detail in the caption would be appreciated.

2. Some fluence values are missing. Fluence 1 in Fig 4 is not given. All fluences for the data in Fig 5 are missing. In particular, as this data should be compared to the data in Fig 4, this is physically mandatory. Denoting the fluence as "high", "medium" or "low" is scientifically insufficient. How are the fluences calculated, as peak fluence or some kind of average? What was the size of the pump laser on the sample, in particular, compared to the sample size?

Reviewer #3:

Remarks to the Author:

The authors added substantial information and clarification in the revised manuscript which significantly improved the readability. The originality of the technique compared to others is clearly stated and its promising application to femtomagnetism will allow unravelling remaining challenges on ultrafast dynamics of spin textured materials.

While the paper does clearly demonstrate a spatial resolution of 16 nm, the discussion about the transverse spin dynamics is not fully convincing. Indeed, the authors are looking at size variation of 10 to 20 nm domain walls, which is at the limit of their maximum spatial resolution, and well below the spatial resolution of 40 nm used in Figure 4. The discussion between lines 169 and 176 needs to be smoothed by making clear that on this Co/Pd systems the technique is at the spatial resolution limit but would be ideally suited to study systems having larger DW like in [12,13]. (Ref 13 is now PRB 102, 174437, 2020).

Altogether, I believe that this work is now close to the high standard of Nature Communications. Therefore I can recommend this manuscript for publication after minor revisions.

Reviewer #1 (Remarks to the Author):

The authors have revised their manuscript to address my concerns. In particular they identify the pump fluence dependent crossover to irreversible changes in the recorded domain pattern. This is an important result that sets the present investigation apart from results using spatially averaged scattering techniques and provides valuable new scientific insight. I therefore support publication of the manuscript in its present form.

We thank the Reviewer for the positive comments and recommending publication in Nature Communications.

Reviewer #2 (Remarks to the Author):

The authors have addressed my comments suitably and I find the manuscript significantly improved. In particular, the part on static modifications of the domain patterns is now well integrated into the presentation of the findings. I still recommend publication in Nature Communications due to the seminal methodical progress combined with important physical findings.

I only have some minor issues remaining that should be addressed before publication:

1. In Fig. 2 d, the meaning of the double-headed arrow is not explained. Should it give a scale between the top and bottom panel? As the bottom panel shows the ratio (rather than the difference), this would not make sense mathematically, in my opinion.

We thank the Reviewer for pointing out this issue. We have revised the figure and caption. We removed the small arrow and left the larger one that highlights the dichroic signal in the complex plane, and note its length to illustrate the strength of the dichroic signal.

Which data points are shown in Fig 2d? Only the ones inside the main FOV? Do the color scales in Fig 2 b,c agree with the axis scaling in panel d top? The caption of Fig 2 still explains panel e as f. More detail in the caption would be appreciated.

Panel **d** shows the data points only within the circular FOV, which is now specified in the caption. The panels **b** (full images) and **d** (scatter plot) are all normalized to the mean complex amplitude in the FOV. The labels to the panels were corrected.

2. Some fluence values are missing. Fluence 1 in Fig 4 is not given. All fluences for the data in Fig 5 are missing. In particular, as this data should be compared to the data in Fig 4, this is physically mandatory. Denoting the fluence as “high”, “medium” or “low” is scientifically insufficient. How are the fluences calculated, as peak fluence or some kind of average? What was the size of the pump laser on the sample, in particular, compared to the sample size?

The pump spot was two orders of magnitude larger than the sample size (approximately 400 μm vs. 3 μm). We specify the nominal incident fluence on the structure, as the local fluence is strongly inhomogeneous on a sub-micrometer scale and affected by scattering at the mask. For the data shown in Fig. 5 we took an image of un-pumped sample to capture the initial state and used pump fluences of 4.0(6) mJ/cm^2 and 1.4(1) mJ/cm^2 for the data marked as high and medium fluence, respectively. For the annealed data, we first increased the pump fluence to the same high fluence (4.0(6) mJ/cm^2) and then gradually decreased the fluence to zero.

We now added the incident fluences to the captions in Fig. 4 and Fig. 5.

We thank the Reviewer again for the revision and for the suggestions that we believe substantially improved our manuscript.

Reviewer #3 (Remarks to the Author):

The authors added substantial information and clarification in the revised manuscript which significantly improved the readability. The originality of the technique compared to others is clearly stated and its promising application to femtomagnetism will allow unravelling remaining challenges on ultrafast dynamics of spin textured materials.

While the paper does clearly demonstrate a spatial resolution of 16 nm, the discussion about the transverse spin dynamics is not fully convincing. Indeed, the authors are looking at size variation of 10 to 20 nm domain walls, which is at the limit of their maximum spatial resolution, and well below the spatial resolution of 40 nm used in Figure 4. The discussion between lines 169 and 176 needs to be smoothed by making clear that on this Co/Pd systems the technique is at the spatial resolution limit but would be ideally suited to study systems having larger DW like in [12,13]. (Ref 13 is now PRB 102, 174437, 2020).

Altogether, I believe that this work is now close to the high standard of Nature Communications. Therefore I can recommend this manuscript for publication after minor revisions.

We thank the Reviewer again for the positive feedback, remarks and the recommendation for publication. As suggested, we have updated the reference list. Furthermore, we have edited the paragraph stating the possible resolution limit to the observation of spin currents. We agree with the Reviewer that the demonstrated resolution is at the limit of what is necessary to fully resolve domain walls in Co/Pd multilayers.

Concerning the suitability for systems with larger domain walls, we have added the reference provided by the Reviewer where we estimate the spatial resolution and domain wall width.

We thank all Reviewers again for the careful consideration of our manuscript and suggestions for the improvements.

Reviewers' Comments:

Reviewer #2:

Remarks to the Author:

The authors have suitably resolved the remaining issues. I do not have any further comments and recommend publication of the manuscript as it is.